Effects of wine-cap Stropharia cultivation on soil nutrients and bacterial communities in forestlands of northern China

Gong Sai
Chen Chen
Zhu Jingxian
Qi Guangyao
Jiang Shuxia jsx6206@163.com
College of Plant Protection, Shandong Province Key Laboratory of Agricultural Microbiology, Engineering Research Centre of Forest Pest Management of Shandong Province, Shandong Agricultural University , Taian , Shandong , China
Buckley Hannah
Electronic publication date: 2018 Oct 9
Publication date: 2018
Volume: 6
Electronic Location ID: e5741
Received 2018 May 2; Accepted 2018 Sep 12
Copyright: ©2018 Gong et al.
Copyright year: 2018
Copyright holder: Gong et al.
License: This is an open access article distributed under the terms of the Creative Commons Attribution License, which permits unrestricted use, distribution, reproduction and adaptation in any medium and for any purpose provided that it is properly attributed. For attribution, the original author(s), title, publication source (PeerJ) and either DOI or URL of the article must be cited.
License URL: https://creativecommons.org/licenses/by/4.0/

Keywords: Stropharia rugosoannulata, Fungal biology, Forest farming, Soil nutrients, Soil microorganisms, High-throughput sequencing

Funding: Agricultural Good Breeds Engineering Project of Shandong Province 2016LZGC026 Spark Science and Technology Demonstration Project of Shandong Province 2015XH019 This work was supported by the Agricultural Good Breeds Engineering Project of Shandong Province (No. 2016LZGC026) and the Spark Science and Technology Demonstration Project of Shandong Province (No. 2015XH019). The funders had no role in study design, data collection and analysis, decision to publish, or preparation of the manuscript.

==============================
Background

Cultivating the wine-cap mushroom (Stropharia rugosoannulata) on forestland has become popular in China. However, the effects of wine-cap Stropharia cultivation on soil nutrients and bacterial communities are poorly understood.

Methods

We employed chemical analyses and high-throughput sequencing to determine the impact of cultivating the wine-cap Stropharia on soil nutrients and bacterial communities of forestland.

Results

Cultivation regimes of Stropharia on forestland resulted in consistent increases of soil organic matter (OM) and available phosphorus (AP) content. Among the cultivation regimes, the greatest soil nutrient contents were found in the one-year interval cultivation regime, and the lowest total N and alkaline hydrolysable N contents were observed in the current-year cultivation regime. No significant differences were observed in alpha diversity among all cultivation regimes. Specific soil bacterial groups, such as Acidobacteria, increased in abundance after cultivation of Stropharia rugosoannulata.

Discussion

Given the numerous positive effects exerted by OM on soil physical and chemical properties, and the consistent increase in OM content for all cultivation regimes, we suggest that mushroom cultivation is beneficial to forest soil nutrient conditions through increasing OM content. Based on the fact that the one-year interval cultivation regime had the highest soil nutrient content as compared with other cultivation regimes, we recommend this regime for application in farming practice. The spent mushroom compost appeared to be more influential than the hyphae of S. rugosoannulata on the soil nutrients and bacterial communities; however, this requires further study. This research provides insight into understanding the effects of mushroom cultivation on the forest soil ecosystem and suggests a relevant cultivation strategy that reduces its negative impacts.

Introduction

The wine-cap Stropharia mushroom (Stropharia rugosoannulata Farlow ex. Murrill) is one of the top ten mushrooms traded internationally and is recommended by the UN Food and Agriculture Organization for export to developing countries (Murrill, 1922; Hawksworth et al., 1996). This mushroom is sciophilous and can be cultivated with different kinds of raw materials, such as straw, sawdust, rice husk and corncobs (CUCEDH, 2013; Domondon & Poppe, 2000; Gong et al., 2016). It is easy to cultivate and can reach a high yield with extensive management. These features make S. rugosoannulata suitable for under-forest cultivation. In practice, this mushroom has been cultivated in large gardens with trees and shrubs (Domondon & Poppe, 2000) and under hardwood shade (Bruhn, Abright & Mihail, 2010). Many experiments have been carried out to increase mushroom production (Bonenfant-Magné, Magné & Lemoine, 2000; Domondon et al., 2004; Bruhn, Abright & Mihail, 2010; Zeng, 2013), which have enabled the large-scale cultivation of S. rugosoannulata.

Cultivating mushrooms in forestlands, including under the shade of nursery stocks, has become popular in China. This kind of mushroom cultivation can efficiently use the large expanses of space under nursery stocks. Meanwhile, the straw by-products, which are usually incinerated or discarded in the field (Lu et al., 2018), can be consumed by the mushrooms, thereby reducing waste and air pollution. Due to this, the Chinese government has encouraged the cultivation of economically valuable mushrooms in forestland. Thus, the wine-cap Stropharia is cultivated under the forest as such a kind of mushroom in several Chinese provinces, including Shandong, Fujian (Zeng, 2013) and Yunnan (Yang et al., 2015).

In China, nursery soil has suffered from improper managements, including flood irrigation and excessive inputs of synthetic nitrogen fertilizer. Additionally, topsoil is removed with seedings and nursery stock transactions each year. All these can cause soil erosion, degradation (Wang, Wang & Sayre, 2004), pollution (Dissanayake & Rajapaksha, 2013) and acidification (Conyers et al., 2011; Geng et al., 2016). Fortunately, the importance of these problems has now become apparent, and several attempts have been made to improve soil conditions (Chadwick et al., 2015; Zheng et al., 2016; Sihi et al., 2017). In several studies, the residual compost waste generated by the mushroom production, i.e., spent mushroom compost, is used in soil bioremediation to improve soil aeration, maintain soil structure (Kadiri & Mustapha, 2010), balance soil nutrient (Uzun, 2004; Jonathan, Lawal & Oyetunji, 2011), and increase soil biological activity (Li et al., 2012). Growing mushrooms under nursey stocks can be a good alternative, as a considerable amount of spent mushroom compost will be left in the soil after mushroom harvesting. However, there is currently very limited understanding of the effects on soil nutrients that are caused by mushroom cultivation. Additionally, how mushroom cultivation will influence microbial community composition is also worthy of attention, given that the hyphae of these mushrooms can select certain bacterial taxa in the soil (Nazir et al., 2010). Finally, there is concern that the cultivation of S. rugosoannulata on forestland might lead to soil nutrient loss (Socolow, 1999). In this study, we investigated how different cultivation regimes affect the sustainable development of S. rugosoannulata stocks under nursery stock shade.

Specifically, we cultivated S. rugosoannulata under nursery stocks in Liying (Jining, Shandong, China), one of the largest centres for seedling production in China. We used four cultivation regimes, based on common methods: (i) fallow for one year after prior cultivation regime (Y010), (ii) two years of continuous cultivation regime (Y011), (iii) current-year cultivation regime (Y001), (iv) one-year interval cultivation regime (Y101), to test the effects of growing S. rugosoannulata on influencing soil nutrients and soil microbial community composition.

Materials and Methods

Experimental site

The experimental forestland was an area of 20 × 150 m, located in Liying Town, Jining City, Shandong Province (116°37′E, 35°30′N, 43 m above sea level). The nursery stock is made up by 7- to 10-year-old trees of horse chestnut (Aesculus chinensis Bunge), which were planted with 2 m spaces between plants in rows and 5 m between rows to achieve a canopy density of 0.7. This location is considered a warm temperate, semi-humid monsoon climate characterized by hot, rainy summers and cold, dry winters, with an annual average temperature of 13.2–14.1 °C. The highest temperature in July exceeded 27 °C, and the annual average temperatures above 10 °C accumulated to 4060.7 °C (growing degree days). The annual precipitation is 650–700 mm, with rainfall from May to August accounting for more than 65% of the total rainfall for the whole year. The soil type was non-calcareous cinnamon tide with a clay loam texture. All these data were obtained from the Jining Soil and Fertilizer Workstations, China (1990).

Sample plots and S. rugosoannulata cultivation

The experimental forestland was divided into five 20 × 30 m grids, which were marked as Y000, Y010, Y011, Y001, and Y101 respectively. Among them, Y010, Y011, Y001, and Y101 were used for mushroom cultivation with different regimes, and each of them was divided into three, 10 × 20 m plots for independent replicates; Y000 was used as a no-cultivation control. The cultivation year of each grid is shown in Table 1.

The cultivation of S. rugosoannulata began in 2013 and was performed every November as described by Gong et al. (2016). The basic materials included 48.9% rice husk, 30% corncobs crushed into particles with a diameter of 0.5 cm–1 cm, 20% sawdust, which was a mixture that contained a variety of hardwood chips, 1% soil acquired from each plot before cultivation and 0.1% lime. These materials were mixed, stacking fermentation was performed, and then distributed onto the sample plots between the plant rows with a thickness of approximately 25 cm. The S. rugosoannulata spawn was divided into blocks of approximately 3 cm in length and inoculated into the fermented material using superimposed square planting. Then, 3 cm of the forest surface soil was sprinkled onto the surface of the fungal bed. The fungal bed was vented and kept moist by a 2–3 cm cover of straw under black plastic film. A micro-spray system was installed in each plot, and the ditch between the cultivation beds drained into a stagnant water well. By April of the next year, fruiting had begun, and by late June, the harvest was complete. The soil was subjected to rotary tillage in November, i.e., the material rotting stage (MRS).

Table 1 Cultivation year of each grid.

Grids	Cultivation year	Description in the text	
	2013	2014	2015		
Y000	0	0	0	No-cultivation control	
Y010	0	1	0	Fallow for one year after prior cultivation regime	
Y011	0	1	1	Two years of continuous cultivation regime	
Y001	0	0	1	Current-year cultivation regime	
Y101	1	0	1	One-year interval cultivation regime	
Notes.

The number “1” in the cultivated year column indicates that the cultivation occurred in the corresponding year, whereas “0” indicates that no cultivation was performed in the corresponding year.

Sample collection and measurements of soil properties

A five-point sampling method was used to collect soil samples in October 2016. The surface organic materials of Y011, Y001, and Y101, and 1 cm of the surface soil of Y000 and Y010 were removed to distinguish the effect of the raw organic materials added from the mushroom cultivation. Five soil cores (5 cm diameter) were collected from each plot with a depth of 30 cm, fully pooled and then sifted using a 2-mm sieve. Subsequently, each soil sample was divided evenly into two portions: one was air dried and used for soil nutrient measurements, and the other was stored at −20 °C before soil DNA extraction.

The soil properties were measured in the Shandong Provincial Key Laboratory of Soil Erosion and Ecological Restoration (Tai’an, Shandong, China). The soil organic matter (OM) content was determined with the potassium dichromate external heating method (Ciavatta et al., 1991). The total nitrogen (TN) content was determined by the dichromate oxidization method (Bremner, 1965). The total phosphorus (TP) content was determined by molybdenum-blue colorimetry after digestion by HF-HClO4 (Jackson, 1958). The alkaline hydrolysable nitrogen (AN) content was determined using the alkaline-hydrolysable diffusion method (Xiong et al., 2008). The available phosphorus (AP) was extracted with sodium bicarbonate and determined using the molybdenum-blue method (Olsen, 1954). The available potassium (AK) was extracted by ammonium acetate and then determined by flame photometry (Carson, 1980). The soil pH was determined according to the international standard with a soil/water ratio of 1:5 (ISO 10390: 2005). The soil field capacity (FC) was measured using the laboratory Wilcox method (Duan et al., 2010).

Soil DNA extraction and polymerase chain reaction (PCR) amplification

The hexadecyl trimethyl ammonium bromide (CTAB) method was used for the soil DNA extraction (Zhou, Bruns & Tiedje, 1996), and the purity and concentration of genomic DNA was monitored by 1% agarose gel electrophoresis. DNA was diluted to 1 ng/µL using sterile water for the PCR. The forward specific primer 515F (5′-GTGCCAGCMGCCGCGGTAA-3′) (Turner et al., 1999) and reverse specific primer 907R (5′-CCGTCAATTCMTTTRAGTTT-3′) (Lane, 1991) were employed to amplify the V4–V5 region of 16S RNA. PCR-based amplifications were performed using Phusion® High-Fidelity PCR Master Mix with GC Buffer and high-fidelity DNA polymerase (New England Biolabs, Ipswich, MA, USA) following an amplification programme of 1 cycle at 98 °C for 1 min, 30 cycles composed of three steps for each cycle (98 °C for 10 s, 50 °C for 30 s, and 72 °C for 30 s), and a final elongation step of 72 °C for 5 min.

Equal volumes of 1× loading buffer (containing SYBR green) and PCR products were mixed and electrophoresed on a 2% agarose gel. Samples with bright main bands between 400 and 450 bp were selected for further experimentation. The PCR products were mixed in equidensity ratios and then purified with a Qiagen Gel Extraction Kit (Qiagen, Hilden, Germany). The library was constructed using TruSeq® DNA PCR-Free Sample Preparation Kit (Illumina, San Diego, CA, USA), and the library quality was assessed on the Qubit@ 2.0 Fluorometer (Thermo Scientific, Waltham, MA, USA) and Agilent Bioanalyzer 2100 systems. The library was sequenced on an Illumina HiSeq 2500 platform at Novogene Bioinformatics Technology Co., Ltd., Beijing, China, and 250 bp paired-end reads were generated. All paired-end reads were deposited in Sequence Read Archive (SRA), BioProject: PRJNA453134.

Bioinformatic analysis

Paired-end reads were assigned to samples based on their unique barcodes and truncated by trimming the barcode and primer sequences. After that, paired-end reads were merged using FLASH (V1.2.7; Magoč & Salzberg, 2011) to obtain raw tags. The raw tags were then subjected to quality filtering using QIIME V1.7.0 (Caporaso et al., 2010) to obtain high-quality clean tags (Bokulich et al., 2013). Default settings (r = 3; p = 0.75 total read length; q = 3; n = 0; Sun et al., 2014) was used for quality filtering. These clean tags were compared with the reference database (Gold database, http://drive5.com/uchime/uchime_download.html) using the UCHIME algorithm (Edgar et al., 2011) to detect and remove chimaera sequences (Haas et al., 2011). Thus, we obtained effective tags. Uparse software (v7.0.1001; Edgar, 2013) was used to assign sequences with more than 97% similarity to an operational taxonomic unit (OTU). Representative sequences that showed the highest frequency for each OTU were screened for further taxonomic assignment. The Mothur method with a threshold of 0.8–1 was selected in QIIME (Version 1.7.0), and the SSU rRNA database (Quast et al., 2013) in SILVA (Wang et al., 2007) was used for taxonomic assignment. To obtain the phylogenetic relationships among different OTUs, multiple sequence alignments were conducted using MUSCLE software (Version 3.8.31; Edgar, 2004). The phylogenetic tree for each sample plot was visualized using GraPhlAn (Asnicar et al., 2015).

The OTU abundance data were rarefied using a standard sequence number corresponding to the sample with the fewest sequences. Subsequent analyses of the alpha diversity and beta diversity were performed based on the rarefied output data. The alpha diversity indices, including Good’s coverage estimator and the Shannon and Simpson diversity indices, were calculated using QIIME (Version 1.7.0). The differences in taxonomic composition were evaluated using a beta diversity analysis. The methods of principal component analysis (PCA), principal co-ordinates analysis (PCoA) and non-metric multi-dimensional scaling (NMDS) were used to illustrate the clustering of different samples. PCA was calculated in the R packages FactoMineR (Lê, Josse & Husson, 2008) and ggplot2 packages (Wickham, 2010), and the Hellinger transformation method (Rao, 1995) was used for PCA. PCoA of the weighted and unweighted UniFrac distances was calculated in the R package “ape” (Lozupone & Knight, 2005). An NMDS of the weighted and unweighted UniFrac distances was calculated according to Peck (2010). A canonical correspondence analysis (CCA) calculated using the R package “vegan” (Oksanen et al., 2007) was used to visualize the relationship between edaphic factors and the bacterial community structure in each sample plot. Prior to performing the CCA, we filtered out the intercorrelated environmental factors that affected sample distribution by using a variance inflation factor (VIF) analysis (Gross, 2003).

Statistical analysis

The soil chemical concentration, dominant taxa and alpha diversity indices were measured, and a one-way analysis of variance (ANOVA) was performed to determine whether differences existed among treatment means at a significance level of α = 0.05. Multiple comparisons were conducted for significant effects using the Tukey’s test at α = 0.05, and FDR of Benjamini & Hochberg (1995) was used for Tukey’s test. These statistical analyses were implemented using the Statistical Program for Social Sciences SPSS (Version 22; IBM, USA).

The linear discriminant analysis (LDA) effect size (LEfSe) (Segata et al., 2011) was used to identify significantly different taxa among groups using the LEfSe software with a default LDA score value of 4. An analysis of molecular variance (AMOVA, Excoffier, Smouse & Quattro, 1992), analysis of similarities (ANOSIM, Clarke, 1993) and permutational multivariate analysis of variance (PERMANOVA or ADONIS, Anderson, 2001) was used to determine differences in the microbial community structure between the groups using the amova function in Mothur software (https://www.mothur.org/). Correlations among edaphic factors with estimated diversity levels were tested for significance via Spearman’s correlations (Algina & Keselman, 1999) performed in R ( R Core Team, 2013).

Results

Soil properties

As shown in Table 2, the cultivation of S. rugosoannulata in forestland changed the soil field capacity, pH, organic matter, total nitrogen, total phosphorus, alkaline hydrolysable nitrogen, available phosphorus and available potassium contents. The ANOVA showed that the soil organic matter and available phosphorus increased significantly in all cultivating regimes of S. rugosoannulata compared with the no-cultivation control. The soil nutrient concentrations in the one-year interval cultivation regime were the highest among all grids. Additionally, the soil total phosphorus and alkaline hydrolysable nitrogen in the fallow for one year after prior cultivation regime, the soil total phosphorus, alkaline hydrolysable nitrogen, and available potassium in the two years of continuous cultivation regime and the soil total nitrogen and alkaline hydrolysable nitrogen in the current-year cultivation regime decreased significantly compared with those of the control. In addition, the soil field capacity and pH in all cultivating regimes were not significantly changed (P < 0.05).

Table 2 Soil properties according to different grids.

Grids	FC	pH	OM (g/Kg)	TN (g/Kg)	TP (g/Kg)	AN (mg/Kg)	AP (mg/Kg)	AK (mg/Kg)	
Y000	18.86 ± 0.029a	6.88 ± 0.033a	8 ± 0.025d	0.44 ± 0.008c	0.41 ± 0.005b	67.85 ± 0.166b	30.59 ± 0.008e	101.12 ± 0.159d	
Y010	19.54 ± 0.035a	6.85 ± 0.048a	9.9 ± 0.125b	0.48 ± 0.004b	0.33 ± 0.002d	39.18 ± 0.263d	40.53 ± 0.026c	119.37 ± 0.088b	
Y011	18.12 ± 0.028a	6.97 ± 0.025a	8.9 ± 0.03c	0.47 ± 0.006bc	0.36 ± 0.002c	58.02 ± 0.157c	53.94 ± 0.191b	94.97 ± 0.176e	
Y001	17.69 ± 0.025a	6.9 ± 0.039a	10.4 ± 0.298b	0.22 ± 0.003d	0.38 ± 0.005b	38.18 ± 0.146e	37.81 ± 0.043d	106.27 ± 0.504c	
Y101	22.72 ± 0.004a	6.88 ± 0.051a	23.4 ± 0.288a	0.95 ± 0.013a	0.57 ± 0.01a	103.13 ± 0.228a	88.29 ± 0.048a	152.1 ± 0.2a	
Notes.

Values are the average of three replicate soil samples. Values followed by the same letter are not significantly different at P > 0.05 (ANOVA, Tukey analysis). The significant maximum and minimum values of each soil property among all grids are shown in bold.

FC soil field capacity

pH soil pH

OM organic matter

TN total nitrogen

TP total phosphorus

AN alkaline hydrolysable nitrogen

AP available phosphorus

AK available potassium

Bacterial community composition

After removing potential chimaeras, a total of 1,127,888 high-quality V4–V5 16S rDNA sequences were analysed across the five grids. These sequences were assigned to 8,751 OTUs. The number of OTUs in the grids ranged from 4,756 to 5,011 (Table S1).

The phylogenetic relationship of different OTUs of each sample was illustrated in Figs. S3–S17. The top ten most abundant phyla represented 94% of the sequences (Fig. 1A), of which, Proteobacteria and Acidobacteria were the most dominant phyla in all groups, representing 55–61% of the total sequences. Among them, only Acidobacteria, Planctomycetes and Gemmatimonadetes showed significant changes in relative abundance between forestlands with one of the cultivation regimes and the no-cultivation control (Fig. 1A, Table S2). Acidobacteria in the current-year cultivation regime was the most abundant, and was significantly more abundant than in the control. The abundance of Planctomycetes in forestland with cultivation was greater than that of the no-cultivation control and was greatest in the one-year interval cultivation regime. Significantly fewer Gemmatimonadetes were observed in the one-year interval cultivation regime than in the other sample groups.

Figure 1 Bar chart of bacterial relative abundance at the phylum level and bar chart based on the LDA value.

(A) Bar chart of bacterial relative abundance at the phylum level. Bar chart based on the LDA value, bacterial community groups in comparison pairs with significant differences (LDA score > 4) in abundance are shown. Comparison pairs: (B) Y010/Y000, (C) Y011/Y000, (D) Y001/Y000, (E) Y101/Y000.

The LEfSe analysis was used to identify the specific bacterial groups in the soil from forestland with the cultivation regimes and in the no-cultivation control. Major differences were observed in the bacterial groups among the samples. Notably, Acidobacteria was the most common group in the soil with cultivation (Figs. 1B–1E). The most frequently observed differences were between the one-year interval cultivation regime and the no-cultivation control (Table S3).

Bacterial α-diversity

Before performing the α-diversity analysis, the OTU abundance data were normalized with a cutoff value of 59,458. In all samples, the Good’s coverage values reached 0.98 (Table S1), indicating that the normalised sequencing data was sufficient to capture the bacterial diversity. The Shannon and Simpson indexes were calculated to evaluate the bacterial diversity (Table S1), and no significant difference was observed among the five grids (p = 0.05), even though slightly lower Shannon and Simpson values were observed in the forestlands with cultivation than in the no-cultivation control.

OTU-level bacterial β-diversity analysis

The PCA of the bacterial community construction in different samples is shown in Fig. 2A. The five treatments were clearly distinguished in the PCA. The first two principal components, PC1 and PC2, best reflected the differences between these treatments and represented variations of 12.35% and 10.04% in the bacterial community, respectively. Within the PC1 axis, the one-year interval cultivation regime was distinct from the other regimes. Within the PC2 axis, the no-cultivation control was distinct. Similarly, the weighted Unifrac-based analysis of PCoA and NMDS (Fig. S1) and unweighted Unifrac-based analysis of PCoA and NMDS (Fig. S2) all showed that the one-year interval cultivation regime was distinct from the other regimes. These data indicated that the bacterial community composition in the one-year interval cultivation regime was relatively distinct from other regimes. However, significant differences in the bacterial community composition were not found between the no-cultivation control and the one-year interval cultivation regime via the AMOVA (Fs = 4.05, P = 0.074), the ANOSIM (R = 1, P = 0.1) or the ADONIS (R2 = 0.52, P = 0.1) (Table S4).

Figure 2 Principal component analysis (PCA) and canonical correspondence analysis (CCA).

(A) PCA based on OTUs of the bacterial community. (B) CCA based on edaphic factors and the bacterial community composition. Different grids are represented by different colours. Spots with the same colour represent the same grids. Edaphic factors are shown as arrows, and the degree of correlation between one edaphic factor and community/species composition is represented by the length of the arrow. Longer arrows indicate higher correlations. The angle constructed by the arrow and the ordination axes indicates the correlation between the edaphic factors and the ordination axes. A smaller angle indicates a higher correlation. FC, field capacity; OM, organic matter; TN, total nitrogen; TP, total phosphorus; AN, alkaline hydrolysable nitrogen; AP, available phosphorus; and AK, available potassium. Available edaphic factors for the bacterial community composition are shown as *, i.e., having a variance inflation factor (VIF) value of less than 20.

The VIF analysis suggested that the soil field capacity, pH, organic matter, total phosphorus, alkaline hydrolysable nitrogen were the uncorrelated edaphic factors in the CCA that could represent the relationship between soil physicochemical properties and bacterial community composition. Based on this model, a total of 60.39% of the variance was explained by CCA1 (24.46%) and CCA2 (19.49%), which were the first two constrained axes of the CCA (Fig. 2B). The CCA suggested that the organic matter, total phosphorus, and alkaline hydrolysable nitrogen were the determinants among the edaphic factors. The correlation analysis showed that only organic matter was significantly associated with the soil bacterial composition (r =  − 0.579, P = 0.024) and the Shannon index (r =  − 0.571, P = 0.026).

Discussion

Effects of cultivating S. rugosoannulata under nursery stock shade on soil properties

Recently, growing S. rugosoannulata under nursery stock shade has been considered a win-win agricultural practice that can improve the quality of nursery stock soil in China. In this study, higher organic matter and available phosphorus content were observed in forestlands with cultivation of S. rugosoannulata compared with the no-cultivation control. However, other soil nutrients did not increase consistently in the forestlands with cultivation; some significantly decreased (Table 2). These results were unexpected because a positive correlation has been observed between the organic matter content and soil fertility (Sadikhani, Sohrabi & Fard, 2014). However, the acute angles between the arrow line representing organic matter content and the arrow lines representing other nutrients in the CCA (TN, TP, AN, AP and AK; Fig. 2B) indicate that organic matter content was positively correlated with the other nutrients in this study also. These results are consistent with those of other studies (Sihi et al., 2017; Zhou et al., 2017).

Cultivation in temperate climates usually results in a significant loss of mineralized organic N in soil (Tiessen, Cuevas & Chacon, 1994). Although farming S. rugosoannulata in forestland is a type of agricultural practice, it is distinct from traditional crop cultivation. The decrease in alkaline hydrolysable nitrogen content in forestland under cultivation (except for the one-year interval cultivation regime) indicated that the following cultivation regimes resulted in nitrogen loss: fallow for 1 year after prior cultivation regime (Y010), two years of continuous cultivation regime (Y011) and current-year cultivation regime (Y001). The one-year interval cultivation regime (Y101) effectively suppressed the nitrogen loss and significantly increased the alkaline hydrolysable nitrogen content. In addition, the one-year interval cultivation regime performed well in maintaining soil fertility and had the highest soil nutrient content (Table 2). In contrast, the two years of continuous cultivation regime resulted in a loss of soil nutrients with a significant decrease in total phosphorus, alkaline hydrolysable nitrogen, and available potassium content (Table 2). The current-year cultivation regime resulted in a significant decrease in total nitrogen and alkaline hydrolysable nitrogen content.

Soil use and management, such as less intensive management, can cause the loss of phosphorus (Leinweber et al., 1999). Herein, total phosphorus loss was found in the fallow for one year after prior cultivation regime, two years of continuous cultivation regime, and current-year cultivation regime. This loss may have resulted from the use of a large amount of water during the fruiting stage of S. rugosoannulata. In contrast, a significant increase in total phosphorus was observed in the one-year interval cultivation regime, indicating that the amount of phosphorus increase was greater than the amount of phosphorus lost in the one-year interval cultivation regime. We hypothesize that spent mushroom compost left after the harvesting of fruiting bodies would add a certain amount of macronutrients like nitrogen, phosphorous and potassium (NPK) (Kim et al., 2011). However, this hypothesis needs to be further tested in the future.

It has been suggested a higher organic matter content may lower the soil pH (Hodges, 1996) and increase the water content at field capacity (Hudson, 1994; Tale & Ingole, 2015). Inconsistent with this, the two years of continuous cultivation regime and the current-year cultivation regime showed an increase of organic matter content, together with a decrease in the field capacity and an increase in the pH. The decreased field capacity may be related to disturbances from farming practices that disrupt the aggregates in the soil structure (Dong, 2017), whereas the increased pH may have resulted from the application of quicklime (Moir & Moot, 2010) on the soil surface before S. rugosoannulata cultivation.

Effects of cultivating S. rugosoannulata under nursery stock shade on the soil bacterial community composition

Using high-throughput sequencing analyses, we observed a consistently higher abundance of Acidobacteria and a consistently lower abundance of Actinobacteria and Firmicutes in the forestlands with cultivation (Table S1). However, reports (Ramirez, Craine & Noah, 2012; Zhou et al., 2017) showed that the abundance of Acidobacteria was reduced with increased nutrient inputs because of the oligotrophic properties of these organisms, and the abundance of Actinobacteria and Firmicutes was increased with increased nutrient inputs because of the copiotrophic properties of these organisms. These inconsistencies may be ascribed to the increased organic matter in forestlands with cultivation that creates an oligotrophic soil environment due to its ability to slowly release nutrients (Tiessen, Cuevas & Chacon, 1994). In addition, we observed a consistently increased abundance of Planctomycetes and Bacteroidetes and a consistently reduced abundance of Chloroflexi and Nitrospirae in our study (Table S2). The further study is needed to reveal the underlying mechanisms for this phenomenon.

Interactions between soil fungi and bacteria are common in nature. For example, fungus-released compounds may impact bacterial selection (Warmink, Nazir & Elsas, 2009; Nazir et al., 2010). During the cultivation cycle of S. rugosoannulata, high-density hyphae are observed in the culture substrate for long periods and are even found in the spent mushroom compost. Additionally, the soil contains a considerable amount of hyphae. Therefore, the changes in bacterial communities in the soil after the incorporation of spent mushroom compost would be consistent with changes in bacterial communities in soil environments that surround the dense fungal hyphae, such as soil microhabitats, i.e., hyphospheres or mycospheres (Johansson, Paul & Finlay, 2004; Nazir et al., 2010), that more or less are densely permeated by the fungal hyphae. In our study, a decrease in bacterial diversity was found in forestlands with cultivation (Table S1), which was consistent with reports showing that the bacterial community diversity is lower than that of bulk soil (Warmink, Nazir & Elsas, 2009; Halsey, De & Andreote, 2016). Additionally, the selection of bacteria by the hyphae of S. rugosoannulata may represent a factor that contributes to the emergence of specific bacterial groups, such as Acidobacteria and Subgroup_6 which also belongs to Acidobacteria in forestlands (Figs. 1B–1E). However, it is possible that spent mushroom compost could be more influential on the soil nutrients and bacterial communities because the hyphae of the wine-cap Stropharia disappear along with the deposition of spent mushroom compost (data not published). Further study is needed to understand the impacts of spent mushroom compost and fungal hyphae on soil texture and microbial communities.

Conclusion

Overall, the increased soil contents of organic matter and available phosphorus and the changes in soil bacterial community composition and diversity in the forestland soil with cultivation suggest that S. rugosoannulata cultivation changed the nursery stock soil properties. Given the positive effects on soil physical and chemical properties of organic matter, the highest contents of soil organic matter in the one-year interval cultivation regime suggested that this regime is most appropriate for forestland soils. In addition, this research suggests that (1) organic matter content is the dominant factor affecting soil bacterial community composition, and (2) the spent mushroom compost after harvesting the fruiting bodies of S. rugosoannulata is important for improving both soil nutrient content and soil bacterial community composition and diversity, due to the more abundant organic matter and hyphae of S. rugosoannulata.

Supplemental Information

Table S1 Alpha diversity index ( α-diversity index) for the bacterial community among different grids

Click here for additional data file.

Table S2 Relative abundance of the top ten dominant phyla among different grids

Click here for additional data file.

Table S3 Number of significantly different bacterial community groups by pairwise comparison between forestlands cultivated with Stropharia rugosoannulata (Y010, Y011, Y001 and Y101) and the no-cultivation control (Y000)

Click here for additional data file.

Table S4 The tests of AMOVA (analysis of molecular variance), ANOSIM (analysis of similarities) and ADONIS (PERMANOVA, permutational multivariate analysis of variance) for comparing bacterial community in each grid

Click here for additional data file.

Figure S1 Weighted Unifrac-based analysis

(A) Principal Co-ordinates Analysis (PCoA). (B) Non-Metric Multi-Dimensional Scaling (NMDS). Different grids are represented by different colors. Spots with the same color represent the same grids.

Click here for additional data file.

Figure S2 Unweighted Unifrac-based analysis

(A) Principal Co-ordinates Analysis (PCoA). (B) Non-Metric Multi-Dimensional Scaling. Different grids are represented by different colors. Spots with the same color represent the same grids.

Click here for additional data file.

Figure S3 The phylogenetic tree of Y000.1 based on the representative sequences of OTUs in Y000.1

The color of the branch represents its corresponding phylum, and each color represents a phylum. The size of the circle is proportional to the abundance of the taxonomic groups. The top 40 taxonomic groups in abundance are represented by solid circles.

Click here for additional data file.

Figure S4 The phylogenetic tree of Y000.2 based on the representative sequences of OTUs in Y000.2

The color of the branch represents its corresponding phylum, and each color represents a phylum. The size of the circle is proportional to the abundance of the taxonomic groups. The top 40 taxonomic groups in abundance are represented by solid circles.

Click here for additional data file.

Figure S5 The phylogenetic tree of Y000.3 based on the representative sequences of OTUs in Y000.3

The color of the branch represents its corresponding phylum, and each color represents a phylum. The size of the circle is proportional to the abundance of the taxonomic groups. The top 40 taxonomic groups in abundance are represented by solid circles.

Click here for additional data file.

Figure S6 The phylogenetic tree of Y010.1 based on the representative sequences of OTUs in Y010.1

The color of the branch represents its corresponding phylum, and each color represents a phylum. The size of the circle is proportional to the abundance of the taxonomic groups. The top 40 taxonomic groups in abundance are represented by solid circles.

Click here for additional data file.

Figure S7 The phylogenetic tree of Y010.2 based on the representative sequences of OTUs in Y010.2

The color of the branch represents its corresponding phylum, and each color represents a phylum. The size of the circle is proportional to the abundance of the taxonomic groups. The top 40 taxonomic groups in abundance are represented by solid circles.

Click here for additional data file.

Figure S8 The phylogenetic tree of Y010.3 based on the representative sequences of OTUs in Y010.3

The color of the branch represents its corresponding phylum, and each color represents a phylum. The size of the circle is proportional to the abundance of the taxonomic groups. The top 40 taxonomic groups in abundance are represented by solid circles.

Click here for additional data file.

Figure S9 The phylogenetic tree of Y011.1 based on the representative sequences of OTUs in Y011.1

The color of the branch represents its corresponding phylum, and each color represents a phylum. The size of the circle is proportional to the abundance of the taxonomic groups. The top 40 taxonomic groups in abundance are represented by solid circles.

Click here for additional data file.

Figure S10 The phylogenetic tree of Y011.2 based on the representative sequences of OTUs in Y011.2

The color of the branch represents its corresponding phylum, and each color represents a phylum. The size of the circle is proportional to the abundance of the taxonomic groups. The top 40 taxonomic groups in abundance are represented by solid circles.

Click here for additional data file.

Figure S11 The phylogenetic tree of Y011.3 based on the representative sequences of OTUs in Y011.3

The color of the branch represents its corresponding phylum, and each color represents a phylum. The size of the circle is proportional to the abundance of the taxonomic groups. The top 40 taxonomic groups in abundance are represented by solid circles.

Click here for additional data file.

Figure S12 The phylogenetic tree of Y001.1 based on the representative sequences of OTUs in Y001.1

The color of the branch represents its corresponding phylum, and each color represents a phylum. The size of the circle is proportional to the abundance of the taxonomic groups. The top 40 taxonomic groups in abundance are represented by solid circles.

Click here for additional data file.

Figure S13 The phylogenetic tree of Y001.2 based on the representative sequences of OTUs in Y001.2

The color of the branch represents its corresponding phylum, and each color represents a phylum. The size of the circle is proportional to the abundance of the taxonomic groups. The top 40 taxonomic groups in abundance are represented by solid circles.

Click here for additional data file.

Figure S14 The phylogenetic tree of Y001.3 based on the representative sequences of OTUs in Y001.3

The color of the branch represents its corresponding phylum, and each color represents a phylum. The size of the circle is proportional to the abundance of the taxonomic groups. The top 40 taxonomic groups in abundance are represented by solid circles.

Click here for additional data file.

Figure S15 The phylogenetic tree of Y101.1 based on the representative sequences of OTUs in Y101.1

The color of the branch represents its corresponding phylum, and each color represents a phylum. The size of the circle is proportional to the abundance of the taxonomic groups. The top 40 taxonomic groups in abundance are represented by solid circles.

Click here for additional data file.

Figure S16 The phylogenetic tree of Y101.2 based on the representative sequences of OTUs in Y101.2

The color of the branch represents its corresponding phylum, and each color represents a phylum. The size of the circle is proportional to the abundance of the taxonomic groups. The top 40 taxonomic groups in abundance are represented by solid circles.

Click here for additional data file.

Figure S17 The phylogenetic tree of Y101.3 based on the representative sequences of OTUs in Y101.3

The color of the branch represents its corresponding phylum, and each color represents a phylum. The size of the circle is proportional to the abundance of the taxonomic groups. The top 40 taxonomic groups in abundance are represented by solid circles.

Click here for additional data file.

Supplemental Information 1 Representative sequences of OTUs

Click here for additional data file.

Supplemental Information 2 Raw data of soil nutrients

Click here for additional data file.

We acknowledge Zhen Liu, Shengming Song from Shandong Agricultural University, China for assisting in the measurement of microbial biomass carbon and chemical properties. We acknowledge Lemei Cao, Haoyu Liu, Lijun Li, and Nianzhao Wang et al. from the college students practice innovation projects of Shandong Agricultural University, China for their help in the measurement of soil physicochemical properties. We also acknowledge the reviewers of this paper for their nice comments.

Additional Information and Declarations

Competing Interests

Author Contributions

DNA Deposition

Data Availability

The authors declare there are no competing interests.

Sai Gong conceived and designed the experiments, performed the experiments, analyzed the data, contributed reagents/materials/analysis tools, prepared figures and/or tables, authored or reviewed drafts of the paper, approved the final draft.

Chen Chen, Jingxian Zhu and Guangyao Qi performed the experiments, approved the final draft.

Shuxia Jiang conceived and designed the experiments, analyzed the data, contributed reagents/materials/analysis tools, authored or reviewed drafts of the paper, approved the final draft.

The following information was supplied regarding the deposition of DNA sequences:

Sequence Read Archive (SRA) BioProject: PRJNA453134.

The following information was supplied regarding data availability:

Figshare: https://figshare.com/articles/16S_forestland_soil_raw_sequence_reads/6207638.

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
