# Peer review of "Effects of wine-cap Stropharia cultivation on soil nutrients and bacterial communities in forestlands of northern China"

_PeerJ, doi:10.7717/peerj.5741_

## Round 0.1 · original submission · Major Revisions

Please pay close attention to the comments from the reviewers on clarifying what was done and, in general, making the paper easier to understand. Please also highlight the significance and scientific contribution of the work.

Reviewer 1 ·

Basic reporting

no comment

Experimental design

no comment

Validity of the findings

no comment

Additional comments

Gong et al. described different regimes of wine-cap Stropharia cultivation on the soil properties and microbial community. They found the wine-cap Stropharia cultivation improved soil organic matter and nutrient availability. No significant difference of microbial diversity was observed but specific bacteria groups increased with wine-cap Stropharia cultivation. It is sounds good topic to culture mushroom in the forestland. However, the MS writing should be greatly improved before it is accepted. First, the introduction section is very poor to state the scientific importance of this study. The statement was more focused on techonoloy question. Second, the description of the materials and methods section is not very clear. What size of the forestland experiment site? The mother materials of each plots is the same? Third, as we known, in the process of mushroom cultivation, a lot of raw organic materials such as straw, wood sawdust were input to the cultivation bed. How to distinguish the effect of the raw organic materials addition from the mushroom cultivation? Moreover, the soil samples collection was clear in this MS. Do the author remove all the raw organic materials from the samples? If not consider this question, the conclusion of the MS should be revised. They found the organic matter determined the soil microbial community. Finally, some interpretation should be clear. For example, the subgoup_6 is not clear enough. Please check the whole MS. The significance of this MS should be highlighted. I can't find any new knowledge of this MS.

Reviewer 2 ·

Basic reporting

Generally the manuscript is clear, however, there are some grammatical errors. I have highlighted a few, but it would take about an hour for a native speaker to check this.

Experimental design

Please see my attached review report.

Validity of the findings

please see my attached review report.

Additional comments

please see my attached review report.

Annotated reviews are not available for download in order to protect the identity of reviewers who chose to remain anonymous.

Reviewer 3 ·

Basic reporting

The structure and English sentences are need to be improved for clear understanding of whole contents of the manuscripts.
Authors addressed the mycosphere (or hyphosphere) effect on the bacterial communities in the forest soil. In introduction, however, cultivation effects (e.g. SMC or cultivating regimes) are only considered. Introduction of possible effect of mushroom hyphae on the bacteria is need to be added.

Experimental design

Experimental design was partially well designed, while some regimes are omitted in the design. For examples, why Y100/Y110/Y111 were not included in the design? If author think it is not needed in this analysis, please provide the reason for exclusion of some regimes.

It is not required, but I feel the lack of analysis. Although differences between all regimes were noted in post hoc analysis, authors mainly focused on difference between control and cultivation plot. I think this is waste of the other results including different effects depend on cultivating regimes. If it is possible to considering the pattern of regime effect, it will improve the quality of the analysis and provide better insights. However, I understand if it is not possible to conduct the additional analysis.

Validity of the findings

The topic is interesting, the most conclusions are based on the evidence provided in the results. However, some conclusion is lack of evidence.
1) “Given the severe nitrogen loss and the low quantity of NIB in Y001, an appropriate application of nitrogen fertilizer is suggested when S. rugosoannulata is first cultivated in forestland”
- Given the results from Y010 or Y011, the nitrogen contents in forest soil were recovered to it of control plot in second year cultivation or non-cultivation. Why additional nitrogen fertilizer is need to apply?

2) “The highest content of soil nutrients in Y101 suggested that overyear cultivation might represent the best regime.”
- Authors mentioned “best regime”, but I cannot understand the meaning of “best”. Best for who? Forest ecosystem or farmer? If it is better for farmer, what do you think the effect of increased nitrogen contents on the forest ecosystem?

Additional comments

The manuscript describes the effect of Stropharia rugosoannulata cultivation on the soil properties and bacterial communities in various regimes. Cultivation of S. rugosoannulata increased organic matters and available phosphorus in forestlands soil. In addition, changed of bacterial communities depend on the cultivation regimes was detected using high-throughput sequencing technology. Given the high interests of the wine cap mushroom in China, this study is important to understand the effect of mushroom cultivation on the forest soil ecosystem, and suggest the possibility of relevant cultivation strategy for reduce its negative impacts. However, several points in the manuscript are need to improve for support author’s arguments.

# Major comments
1. Many part of analysis methods or results are lack of detail information or misapplied. This is severe problem, and should be fixed appropriately. Details were noted on the below.

2. Authors used “mycosphere” concept, while it can be inappropriate. Of course, mycosphere is sometime used broad meaning that the environments influenced by the fungus or mushroom. However, mycosphere is more largely used as the soil environments under the fruiting body (mushroom). Thus, Stropharia cultivation plot is not mycosphere, but can be called different concept such as hyphosphere. The previous studies for hyphosphere, shiro of T. matsutake, or ectomycorrhizal mat are agree with the results from this study. Therefore, I recommend that change the mycosphere concept to hyphosphere or add the studies for hyphosphere effect on the bacterial additionally.

3. I suggests additional analysis to detect pattern of bacterial communities. Bacterial communities were separated too much depend on cultivation regime. It can be true, but I worried the analysis miss real pattern of nature. Authors used PCA, but it can be artificial patterns can be generated if OTU matrix were not transformed well. Distance-based ordination analysis (e.g. PCoA or NMDS) may be other options. In addition, weighted Unifrac-based analysis can elucidate different pattern of bacterial communities that was missed by non-phylogenetic distance based analysis.

4. What do you think which factor is more influential on the soil nutrients and bacterial communities between effect from SMC and mushroom hyphae?

# Minor comments
Line 1: In title, “bacterial properties” is inappropriate in this context. I think “bacterial communities” is better.
L32: Is AP meaning available potassium or available phosphorus?
L36: Subgroup_6 is lacking information. Where this taxa is belonging?
L50: What is the meaning of “rich raw materials”, specifically for what kind of material?
L57: Change “N” to “nitrogen”, and provide citation.
L74: How incineration or discard reduce pollution?
L95: What is the meaning of “excellent”?
L118: Change “soil measurements” to “measurements of soil properties”.
L122: Where is the results from the isolation?
L125-126: Authors can change “the Shandong Provincial Key Laboratory of Soil Erosion and Ecological Restoration, Tai’an, Shandong, P. R. China” to “the Shandong Provincial Key Laboratory of Soil Erosion and Ecological Restoration (Tai’an, Shandong, China)”.
L130: N means nitrogen?
L141-143: Add the appropriate reference of the primers.
L144: Add “ribosomal”.
L148: Add “final”.
L159: Change “Bioinformatics” to “Bioinformatic”.
L163: Change “Qiime” to “QIIME”. What option did you use for quality filtering?
L164-165: What kind of the reference database was used? Gold databased from Broad Microbiome Utilities or SILVA? Please provide the citation.
L169: Change “annotation” to “taxonomic assignment”, and “species annotation” to “species assignment”.
L170: What the meaning of Mothur method? Naïve Bayes classifier?
L171: Change “SSUrRNA” to “SSU rRNA”. The citation, Quast et al. 2013, is not present in the reference list.
L172: The purpose of sequence alignment is construction of distance matrix or phylogenetic tree, not for detecting community difference. What kind of method did you use?
L174: Change “normalized” to “rarified”.
L178: QIIME
L179: Did you transform the OTU matrix before conducting PCA?
L180: please give the citation.
L181: Is the citation, Sheik et al. 2012, relevant?
L187: What kind of multiple test correction method was used?
L200-201: please match the order of soil properties to the order in Table 2.
L205: please match the order of plot groups to the order in Table 2.
L245: VIF analysis is for identifying the multicollinearity in the soil properties, but is not for detecting contribution of soil properties on the bacterial communities. Author may be confused on the VIF and CCA analysis.
L294-306: As mentioned above, change or add add hyphosphere concept
L311: What is meaning of NIB?
L314: What is the evidence that show OM content was the primary factor? CCA showed most of soil factors significantly influence on the bacterial communities.
Table 2, L3: Change “AMOVA” to “ANOVA”.
Figure 1: please give more information on the subgroup_6. I don’t think that “unidentified Acidobacteria” is inappropriate taxonomic group because it is complexity of various taxa that is difficult to identify based on sequence database, thus there is no taxonomical or ecological meanings.

---

## Round 0.2 · Minor Revisions

Please make the remaining minor changes suggested.

Reviewer 1 ·

Basic reporting

no comment

Experimental design

no comment

Validity of the findings

no comment

Additional comments

This version of the MS has been improved according to the suggestions.

Reviewer 3 ·

Basic reporting

I cannot fully understand the meaning of “best regime for soil” in the manuscript, and it might be due to lack of connection between background information and results from this study because the information is scattered in the manuscript. Based on Introduction and author’s responses, it seems that soil improvement is needed in China and mushroom cultivation in forestland may help to increase soil nutrients. If my understanding is right, please clarify the rationale of this study based on the connection between soil condition in China and the possible effect of Stropharia cultivation on soil nutrient, and please specify the criterion for “best regime” (e.g. increasing OM content in soil) in objectives parts.

Experimental design

Although authors added the details, many parts are need to add more information and analysis.

L156-158: Primer 515F and 806R were not developed by Zhang et al., (2014). Please give appropriate citation.
L186-188: Please give a taxonomic assignment method. Mothur is not the method. Based on the reference (Wang et al., 2007), authors might use Naïve bayesian classifier.
L188-189: Did you make a phylogenetic tree? If you made, please give the information.
L204-205: What kind of multiple testing correction method was used for Tukey’s test? Bonferroni or FDR of Benjamini-Hochberg?
L209-211, 265-267: AMOVA based on sequence alignment (or matrix) is not enough to test community differences (beta-diversity) because it only consider sequence difference while difference of OTUs composition is missing. Thus, I suggest add MANOVA-based test (e.g. ADONIS, PERMANNOVA, ANOSIM) for comparing bacterial community in each regimes.
L213: What kind of transformation method was used for PCA? Hellinger?

Validity of the findings

None

Additional comments

The manuscript has been revised well, while some parts are need to be changed more.

# Major comments
1. Authors discussed about N loss in second paragraph. However, TP loss was also significant. Please discuss about P loss as did for N loss.

# Minor comments
Line 40: The meaning of “beneficial to the forest soil” is ambiguous. Please, specify it (e.g. beneficial to nutrient condition in the forest soil).
L69: Change “soil” to “soil nutrient”.
L75: Change “pollution” to “air pollution”.
L83-84: Sentence is not complete.
L98: Change “20m × 150m” to “20 m × 150 m”.
L194-200: This part is overlapped to L213-L219. Please merge it.
L223: Remove “and” between pH and OM.
L238: These phyla were not significantly different between “all cultivation regime” and non-cultivation regime. Thus, change “cultivation” to “one of cultivation”.
L249: “diversity” is need to be in non-italic.
L273-L275: Correlation between OM and the number of OTUs or Shannon’s diversity is not mean that OM is main driver on bacterial “community”. It is the result for relationship between OM and alpha diversity. Of course, it seems that OM influence on both of diversity and community structure, but authors need to write the sentence appropriately.
L321: Change “fingi” to “fungi”.
L342: “best regime”. Same above.
Figure 1: Can you change color of Y010 and Y011 to green for consistent to Y001 and Y101?

---

## Round 0.3 · Minor Revisions

Thanks for your comprehensive responses to the reviewer comments. I would like you to make two more changes:
1. Please add a comment to the manuscript about the phylogenies that you have generated. If the reviewer has a question about this, other readers were too, so you need a response to that question in the manuscript. You can either add the information as additional supplementary files, and/or add some text to the methods and results.
2. Please edit the manuscript for grammar.

---

## Round 0.4 · Minor Revisions

Thank you for addressing my previous comments. I recommend a simpler title: "Effects of Wine-cap Stropharia Cultivation on Soil Nutrients and Bacterial Communities in Forestlands of Northern China". Also, I have further edited your manuscript for grammar and understandability. I suggest that you remove all acronyms wherever possible. I also strongly recommend that you remove the letter-number codes for the treatments and give them short, descriptive names that you define clearly in the introduction. This will make the manuscript even easier to understand. I have attached some comments here in tracked changes that I hope you will find helpful with making these changes. Unfortunately, I can only attach a PDF version of the file here; please email me directly if you want the Word doc version (Hannah.Buckley@aut.ac.nz).

---

## Round 0.5 · accepted · Accept

Thank you for responding to all my final comments. I am looking forward to seeing your article in print!

#